# Smooth Stitching Method for the Texture Seams of Remote Sensing Images Based on Gradient Structure Information

**Danjun Deng**

Computer Science Department, Hubei PolyTechnic University, Huangshi 435002, China;
danjun_deng@126.com or 207018@hbpu.edu.cn

**Abstract:** Traditional smooth stitching method for the texture seams of remote sensing images is affected by gradient structure information, leading to poor stitching effect. Therefore, a smooth stitching method for the texture seams of remote sensing images based on gradient structure information is proposed in this research. By matching the feature points of remote sensing images and introducing a block link constraint and shape distortion constraint, the modified stitching image is obtained. By using remote sensing image fusion, the smooth stitching image of texture seams is obtained, and the local overlapping area of the texture is optimized. The main direction of texture seams is determined by calculating the gradient structure information of texture seams in horizontal and vertical directions. By selecting the initial point, the optimal stitching line is extracted by using the minimum mean value of the cumulative error of the smooth stitching line. By using the method of boundary correlation constraints, matching the feature points of the texture seams of remote sensing images and selecting the best matching pair, a smooth stitching algorithm for the texture seams of remote sensing image is designed, which realizes the smooth stitching of the texture seams of remote sensing images. Experimental results show that the design method has good performance in stitching accuracy and efficiency in the smooth stitching of the texture seams of remote sensing images. Specifically, the Liu et al. and the Zhang et al. methods that are the benchmark studies in the literature are introduced as a comparison, and the stitching experiment is carried out. The test is carried out according to accuracy and time and the proposed method achieves better results by almost 25%.

**Keywords:** gradient structure information; remote sensing image; texture seam; smooth stitching

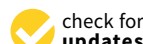



## 1. Introduction

The development of image stitching technology has gone through many stages. The earliest image stitching technology mainly used the principle of gray-scale symbiosis to obtain the image similarity change matrix through gray to complete the image stitching [1]. Later, more and more texture stitching methods for remote sensing images emerged, among which the more prominent was the research method based on scale-invariant feature transform (SIFT) feature points. This kind of method used SIFT feature points to rotate the remote sensing image to obtain the gradient structure information. The image stitching was completed through the reorganization of gradient information feature points [2]. However, traditional remote sensing image stitching methods based on SIFT feature points can no longer complete the stitching of remote sensing images as high-resolution remote sensing images' textures contain a lot of information. This research proposes a smooth stitching method for the texture seams of remote sensing images based on gradient structure information, which is of great significance for realizing high-resolution image stitching [3].

Liu et al. [4] proposed a smooth stitching method of the texture seams of remote sensing images based on Oriented FAST and Rotated BRIEF (ORB) features to solve the problem that high-resolution unmanned aerial vehicle (UAV) remote sensing images could not be stitched by the traditional scale-invariant feature transform (SIFT) algorithm.

First, the remote sensing images that needed image stitching were selected, and the ORB algorithm was used to obtain the feature vectors of the images. Then the remote sensing image's threshold was obtained by the neighbor algorithm. After that, fine matching was performed on the feature points. Finally, the weighted average algorithm was used to perform smooth stitching on the texture seams of remote sensing images. Compared with the traditional SIFT algorithm, this method improved the processing speed of remote sensing image stitching by more than ten times, and the processing effect was better. The Zhang et al. method [5] proposed an image stitching method to improve the best stitching line for the phenomenon that the texture seams of moving objects could not be smoothly stitched. First, the ORB algorithm was used to extract the texture feature points of the remote sensing images. Then, the trapezoid matrix was arranged based on the feature points to remove the incorrectly matching items. Then, the difference of the processed image was corrected, and the improved energy function was used to calculate the stitching function. Finally, the feature point information of the texture seams was subjected to weight analysis, and the images were fused based on the result of the weight analysis. This method could be used for the stitching of high-resolution images.

Through various registration methods, the overlapping areas of two images can be well aligned, and the non-overlapping areas usually have serious distortions. Chang et al. [6] proposed the SPHP algorithm, which corrects the shape of the stitched image, and reduces the projection distortion. Lin et al. [7] proposed a homography linearization method, which is also a shape correction problem; the natural appearance of the stitching results is improved with this method by a level compared with SPHP. Chen et al. [8] proposed the global similarity prior (GSP) to obtain natural stitching results, which uses line alignment constraints to set constraint terms. Liao et al. [9] used two single-perspective warps to register images to produce natural stitching results. Li et al. [10] proposed a novel quasi-homography to solve the line blending problem between the homography transformation and the similarity transformation by linearly scaling the horizontal component of the homography to create a more natural panorama.

Additionally, an UAV image mosaic usually needs additional information, such as camera calibration parameters, and position and rotation data from the GPS/IMU, ground control points or a reference map to achieve accurate mosaic results. Xue et al. [11] proposed a novel image mosaic method without the requirements of camera calibration parameters, camera poses, or any 3D reconstruction procedure. This method can obtain visually satisfactory mosaic results automatically using only the raw 2D imagery captured from an UAV. Liu et al. [12] considered mosaic for low overlap UAV aerial images and found that the commonly used methods and software were not suitable for this type of data. Then, a novel adjustment model for mosaic low overlap sweeping images was proposed. Cai et al. [13] proposed a robust transform estimator based on residual analysis and its application to UAV aerial images. Estimating the transformation between two images from the same scene is a fundamental step for image registration, image stitching and 3D reconstruction.

However, none of the above methods consider the gradient information of images, which leads to poor stitching effects for remote sensing images. For this reason, this paper proposes a smooth stitching method for the texture seams of remote sensing images based on gradient structure information.

## 2. Design of Smooth Stitching Method of the Texture Seams of Remote Sensing Images

### 2.1. Local Optimization and Fusion of the Texture Overlapping Area of Remote Sensing Images

In order to optimize and fuse the local features of the texture overlapping area of remote sensing images, and prevent distortion in the stitching process, visual communication technology is used to perform finite element analysis on remote sensing images [14]. The finite element meshes are transformed into similar textures. The texture overlapping area of the transformed remote sensing images is locally distorted. Taking a triangular remote

sensing image to be stitched as an example, we set three transformation points as $\overline{V}_1$, $\overline{V}_2$, $\overline{V}_3$, and the relationship between the vertex $\overline{V}_1$ and the other two points is as follows:

$$\overline{V}_1 = \overline{V}_2 + u(\overline{V}_3 - \overline{V}_2) + v(\overline{V}_3 - \overline{V}_2) \tag{1}$$

where the coordinates in the stitching texture coordinate system of the remote sensing image from the vertex $\overline{V}_1$ to $\overline{V}_1$ and $\overline{V}_3$ are $u$ and $v$. If the local remote sensing image texture image $\hat{I}_2$ to be optimized and the similarly changed image $\overline{I}_2$ are not matched and transformed, the overlapping position of $V_1$ and $V_2$, $V_3$ cannot be calculated, then the similar texture change formula is:

$$E_S(\overline{V}_i) = w_s \|\overline{V}_1 - (\overline{V}_2 + u(\overline{V}_3 - \overline{V}_2) + v(\overline{V}_3 - \overline{V}_2))\|^2 \tag{2}$$

where $w_s$ is the remote sensing image transformation weight. By calculating the local distortion coefficient $E_S(\overline{V}_i)$ of similar texture transformation in the finite element grid [15], the local distortion energy formula of the texture overlapping area of transformed remote sensing image is obtained.

When using visual communication technology to optimize and fuse the parts of the texture overlapping area of remote sensing images [16], only the texture of the overlapping area of a single grid that needs to be optimized is locally similarly transformed, such as the remote sensing image $I_1$ that needs to be locally optimized and fused. In the process of local similarity transformation, it is necessary to refer to the texture attributes of the overlapping area of the image $\hat{I}_2$, and then optimize the transformation of each coordinate point in the horizontal coordinate system and the vertical coordinate system and perform local distortion constraints on the optimized transformed grid links to make its fusion degree close to zero. At this time, it means that the texture overlapping area of the two remote sensing images has the same local optimization. The least squares method is used to calculate the local fusion optimal solution [17], and the horizontal constraint term of the remote sensing image texture is obtained:

$$E_{l_1}\{N_5\} = s_{l_1}\left(\left|\left(2\widetilde{V}_6 - \widetilde{V}_5\right) - \left(2\widetilde{V}_7 - \widetilde{V}_6\right)\right|^2\right) \tag{3}$$

where $s_{l_1}$ is the sum of the remote sensing image texture grid block $N_5$ to be stitched and the adjacent grid block that needs to be fused. Knowing the vertices $\widetilde{V}_6$, $\widetilde{V}_5$ and $\widetilde{V}_7$ in the remote sensing image, the following formula is used to obtain the vertical constraint term of the remote sensing image texture:

$$E_{l_2}\{N_5\} = s_{l_2}\left(\left|\left(2\widetilde{V}_8 - \widetilde{V}_6\right) - \left(2\widetilde{V}_{10} - \widetilde{V}_8\right)\right|^2\right) \tag{4}$$

where $s_{l_2}$ is the sum of the remote sensing image texture grid block $N_5$ to be stitched and the lower grid block that needs to be stitched. Knowing the vertices (8, 6 and 10) in the remote sensing image, the following formula is used to obtain the diagonal constraint term of the remote sensing image texture:

$$E_{l_3}\{N_5\} = s_{l_3}\left(\left|\left(2\widetilde{V}_{11} - \widetilde{V}_6\right) - \left(2\widetilde{V}_{15} - \widetilde{V}_{11}\right)\right|^2\right) \tag{5}$$

where $s_{l_3}$ is the product of the remote sensing image texture grid block $N_5$ that needs to be fused and the grid block to be stitched with connection relationship, (11, 6 and 15), are the vertices in the known remote sensing image.

The energy formula $E_S$ and grid energy formula $E_l$ of the remote sensing image texture constraint term are mixed in operation [18] to realize the local optimization of the remote sensing image texture:

$$E = \alpha E_S + \beta E_l \tag{6}$$



where $\alpha$ and $\beta$ are the weighted thresholds. After fusing the locally optimized remote sensing images, we can finally obtain the stitched image of the texture overlapping area of remote sensing images.

By matching the feature points of remote sensing images, block link constraints and shape distortion constraints are introduced to obtain the corrected stitched image. The remote sensing image fusion method is used to obtain a smooth stitched image of the texture seams of remote sensing images, which optimizes the texture of the local overlapping area of remote sensing images.

### 2.2. Extraction of the Optimal Stitching Line of the Texture Seams of Remote Sensing Images

The main direction line of the texture overlapping area of the remote sensing images is divided into a vertical direction line and a horizontal direction line. If $I_s$ represents the initial remote sensing image and $I_t$ represents the target image, then the main direction line of the overlapping area is $I = I_s + I_t$. The principle of gradient structure information is used [19] to calculate the gradient structure information in the horizontal and vertical directions, namely:

$$S_1 = \begin{bmatrix} -1 & -2 & -1 \\ 0 & 0 & 0 \\ 1 & 2 & 1 \end{bmatrix}, S_2 = \begin{bmatrix} -1 & 0 & 1 \\ -2 & 0 & 2 \\ -1 & 0 & 1 \end{bmatrix} \tag{7}$$

According to the above formula, the main direction of the texture seams of the remote sensing images can be determined, as shown in Figure 1.

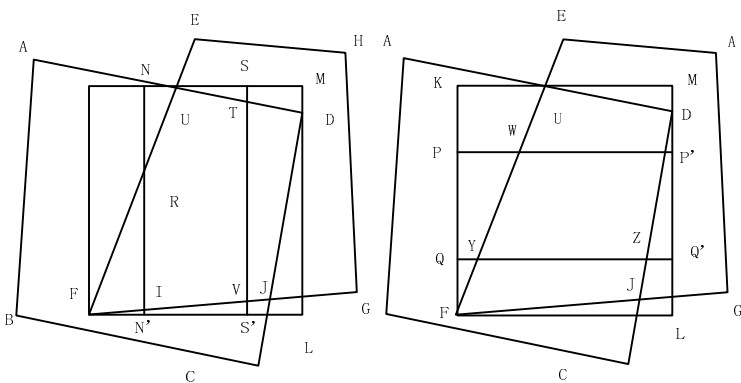

**Figure 1.** Schematic diagram of the main directions of the texture seams of remote sensing images.

By calculating a set of initial points and then deriving them as a starting point, the optimal stitching line is selected from the smooth stitching lines of the last multiple texture seams of remote sensing images, thereby saving the calculation time [20]. Taking Figure 1 as an example, we select five candidate initial points on NS. The coordinates of point N, point U, and point S are $(X_N, Y_N)$, $(X_U, Y_U)$ and $(X_S, Y_S)$, respectively. However, because the ordinate values of the three candidate initial points are the same, the coordinate corresponding to the point on NS is $(X_{pn}, Y_{pn})$, the line from the point $P_n$ to the intersection point of UR and UT is denoted as $q_n$, and the gradient error can be calculated by:

$$C(q_n) = (1 - \beta)C_g(q_n) + \beta C_d(q_n), 0 < \beta < 1 \tag{8}$$

where $\beta = 0.4$, $C_g$ represents the smoothness of the gradient, and $C_d$ represents the similarity.

Using the error calculation formula [21] between the gradient smoothness and the similarity, the gradient error value of the texture seams of remote sensing images is calculated. The following formula is used to divide the five candidate initial points on NS into five groups, namely:

$$\psi_m = \{q_n, n = (m-1)\lambda + 0, (m-1)\lambda + 2, \cdots, m\lambda\} \tag{9}$$

$g_m$ is used to mark the smallest point in each group, the number of smooth stitching lines of the texture seams of remote sensing image is $m = 1, 2, 3, 4, 5$, and five initial points are calculated.

According to the calculated five smooth stitching lines of the texture seams of remote sensing images, the minimum cost is the optimal stitching line of the texture seams of remote sensing images [22]. Assuming that the corresponding coordinate of the point $V_c$ is $(X_c, Y_c)$, one of the three candidate initial points is selected as the next growth point.

According to the calculated cumulative gradient error, the iteration point is determined as $V_n$. Let the cumulative gradient errors of the three candidate initial points be $A(V_1)$, $A(V_2)$ and $A(V_3)$, respectively, then the error calculation formula for each candidate initial point is:

$$\begin{cases} A(V_1) = \sum_{j=1}^{7} C(V_{ej}) \\ A(V_2) = \sum_{j=2}^{8} C(V_{ej}) \\ A(V_3) = \sum_{j=3}^{9} C(V_{ej}) \end{cases} \tag{10}$$

where the number of neighboring points of the candidate initial point is $j = 1, \cdots, 9$, $A(V_1)$ represents the cumulative point composed of the point $V_1$ and six adjacent points. If $(X_c - 4 + j, Y_c + 1)$ is the coordinate of the point $V_{ej}$, then the point with the smallest cumulative value can be selected according to the above formula. It can be obtained from the above formula that when the accumulated values of two points are equal, the priority is sorted as $V_{n2}$, $V_{n1}$ and $V_{n3}$.

Therefore, the conditions for the selection of candidate points of the point $V_n$ are deduced and expressed as:

$$V_n = \begin{cases} V_{n2}, A(V_{n2}) > \min\{A(V_{n1}), A(V_{n3})\} \\ V_{n3}, A(V_{n3}) < \min\{A(V_{n1}), A(V_{n2})\} \\ V_{n1}, A(V_{n3}) = \min\{A(V_{n1}), A(V_{n2})\} \end{cases} \tag{11}$$

According to the obtained five smooth stitching lines of the texture seams of remote sensing images, it is supposed that the $m$th smooth stitching line of the texture seams of remote sensing images contains $j_m$ points, and the coordinate of the $j$th point is $V_n(m, j)$, then the mean value of the minimum cumulative error of the smooth stitching lines can be calculated by:

$$L(m) = \sum_{j=1}^{j_m} A(V_n(m, j)) / j_m \tag{12}$$

The mean value $L(m)$ of minimum cumulative error calculated in the above formula is the optimal stitching line of the texture seams of remote sensing images, which can smoothly pass through the remote sensing image and optimize the removal effect of the smooth stitching line of the texture seams of remote sensing images [23].

By calculating the gradient structure information of the texture seams of remote sensing images in the horizontal and vertical directions, the main direction of the texture seams is determined, the initial point is selected, and the mean value of the minimum cumulative error of the smooth stitching line is used, and finally, the optimal stitching line is extracted.

### 2.3. Design of a Smooth Stitching Algorithm for the Texture Seams of Remote Sensing Images

It is supposed that the texture feature point cluster after the local optimization of the remote sensing image texture overlapping area is $(A_i, B_i)$, $A_i$ is on the remote sensing image texture $L$, $B_i$ is on the remote sensing image texture $R'$. Then, we can calculate the

distance from any feature point $A_K$ to the feature point $L$ of remote sensing image texture, and the texture local feature vector of $A_K$ and $B_K$, as follows:

$$\begin{cases} DA_K = (\overline{A_1}\overline{A_K}, \overline{A_2}\overline{A_K}, \cdots, \overline{A_i}\overline{A_K}) \\ DB_K = (\overline{B_1}\overline{B_K}, \overline{B_2}\overline{B_K}, \cdots, \overline{B_i}\overline{B_K}) \end{cases} \tag{13}$$

We then calculate the relationship function between two local feature vectors of remote sensing image textures:

$$R(DA_K, DB_K) = \frac{\overline{A_1}\overline{A_K} \times \overline{B_1}\overline{B_K} + \cdots + \overline{A_i}\overline{A_K} \times \overline{B_i}\overline{B_K}}{\sqrt{\overline{A_1}\overline{A_K}^2 + \cdots + \overline{A_i}\overline{A_K}^2}\sqrt{\overline{B_1}\overline{B_K}^2 + \cdots + \overline{B_i}\overline{B_K}^2}} \tag{14}$$

It is supposed that the matching threshold for remote sensing image texture is $\varepsilon$, and the texture feature points larger than this threshold have better similarity and transformation, so this point is the best texture stitching point. In the process of stitching remote sensing image textures, texture feature points are the core elements of stitching [24]. Through the spatial summation method, the distribution of each feature point in the remote sensing image texture is described. The stitching of the remote sensing image textures is completed through the fusion of the texture feature points. This method does not consider the sequence relationship, so the result is not accurate enough [25]. For the problem that the remote sensing image cannot realize the smooth stitching of texture seams, a new texture stitching method is proposed. This method uses $v_i$ as a feature transformation reference point and rebuilds a new texture feature point matrix according to the order relationship of the remote sensing image textures [26].

The new texture stitching method has the same principle as the initial processing method, and the coordinates of the reference point $v_i$ for feature transformation are obtained:

$$l(v_i, v_j) = l_{i,j} = (v_i, v_j), j = 1, 2, \cdots, N; j \neq i \tag{15}$$

According to the texture natural relationship in the remote sensing image texture feature points (Equation (14)), we change $l_{i,j}$ to matrix $L_i$:

$$L_i = (l_{i,i+1}, l_{i,i+2}, \cdots, l_{i,N}, l_i, \cdots, l_{i,i-2}, l_{i,i-1})^T \tag{16}$$

The above formula is a new remote sensing image texture description. Compared with the initial remote sensing image texture description, the new method has a more correct sequence relationship, and can accurately see all the information points of the remote sensing image texture without performing multiple segmentation calculations on a 2D image, which improves the stitching efficiency [27].

The new remote sensing image texture description is processed uniformly to ensure that the original feature vector remains unchanged, and the feature points of the remote sensing image texture are arranged according to the description $L_i$ to form a trapezoidal structure information matrix $(N-1) \times N$:

$$A = A(V) = [L_1 L_2 L_3 \cdots L_{N-1} N] \tag{17}$$

Unified processing is performed on each row of the trapezoidal structure matrix:

$$L_i^{(j)} = \frac{l_i^{(j)}}{\max_{i=1,2,\cdots,N} \|l_i^{(j)}\|} \tag{18}$$

where the structure information point in the trapezoidal structure matrix $\|l_i^{(j)}\| = \sqrt{x_j^2 + y_j^2}$ is $l_{i,j}$. Through the processing in Equation (17), the new remote sensing image texture stitching method has better fusion effect.

In order to improve the stability of the proposed method, the texture seams are smoothly stitched:

$$L_i = \left( l_i^1, l_i^2, \cdots, l_i^{N-1}, \right)^T \tag{19}$$

where the sequence $N - 1$ can be split into several mutually restricted trapezoidal structure sequences $[1, t]$, $[t + 1, 2t]$ and $[2t + 1, 3t]$, where $t$ is the smooth stitching coefficient of texture seams. Weighted analysis is performed on each sequence, respectively, to find the average value:

$$g_i^{(j)} = \frac{1}{t} \sum_{c=(r-1)_{t+1}}^{ct} l_i^{(j)} \tag{20}$$

where $c = 1, 2, \cdots, W$, $c$ is the sequence number of the $c$th ladder structure in the sorted ladder structure sequence. We have $W = (N - 1)/t$.

After feature extraction and processing of remote sensing image textures, the remote sensing image textures to be stitched have corresponding texture transformation points, and the minimum variance calculation is used to obtain the best stitching point of remote sensing images [28].

Taking the central feature point of the remote sensing image texture as the reference point of similar transformation, the coordinate transformation formulas of A and B can be obtained as:

$$X = [1, X', Y'] \begin{bmatrix} \alpha_0 \\ \alpha_1 \\ \alpha_2 \end{bmatrix}, Y = [1, X', Y'] \begin{bmatrix} \beta_0 \\ \beta_1 \\ \beta_2 \end{bmatrix} \tag{21}$$

where $(X', Y')$ and $(X'', Y'')$ are the coordinates of the reference point of similar transformation in A and B, respectively. Equation (21) can be solved by a linear equation set in three unknowns, and the equation only needs to be solved for the transformed reference point; finally, the stitching situation of the remote sensing images is verified.

The texture feature point fusion of remote sensing images is the final work in the entire image stitching process. The smooth fusion of the texture seams is carried out by using the gradient structure information transformation method. Generally, images are divided into original remote sensing images and improved remote sensing images. The improved remote sensing image is rotated to obtain the rotation angle, and the improved remote sensing image after the rotation is overlapped with the original remote sensing image, and the effect of the smooth stitching is examined. The specific operation is shown in Figure 2.

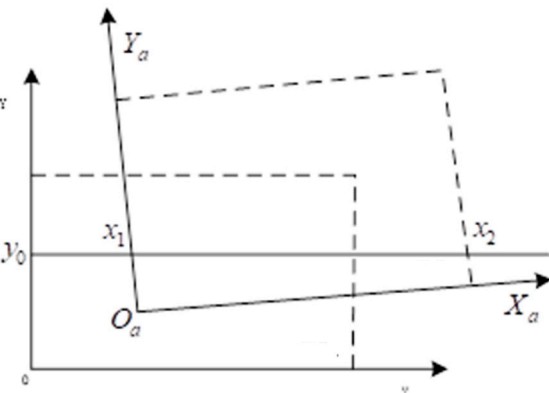

**Figure 2.** The fusion process of remote sensing images.

The coordinate system XOY and the coordinate system $X_a O_a Y_a$ are the coordinate systems of the remote sensing image one and the remote sensing image two, respectively. The start and end points of the remote sensing image two are obtained by the method of line segment cutting, so that the relationship between the overlapping area and the

point can be calculated after the coordinate conversion. Then the difference method is used to obtain the converted pixel color of the remote sensing image, and finally the linear transition is used to fuse the stitched remote sensing images. In summary, the boundary correlation constraint method is used to match the feature points of the texture seams of remote sensing images, the best matching pair is selected, and a smooth stitching algorithm is designed for the texture seams of remote sensing images to achieve the smooth stitching of the texture seams.

## 3. Experiment and Results Analysis

In order to verify the effectiveness of the smooth stitching method of the texture seams of remote sensing image based on gradient structure information in the application, simulation experiments needed to be designed. The operating system of the simulation experiment selected a Windows 7 system computer with 8 GB memory and 3.5 GHz frequency, and used MFC to write the simulation program. The calculation operation of the experiment was completed by the MATLAB compiler to convert the simulation program into a file that can be executed.

### 3.1. Experimental Setup

During the experiment, three groups of different remote sensing images were selected as the test objects of the simulation experiment. The remote sensing images used in Experiment 1 and Experiment 2 were from two remote sensing images with overlapping areas from different locations. The remote sensing images of Experiment 2 were set to 30 degrees. Rotation angle, the remote sensing image in experiment 3, contained a 5 dB low-noise image.

The Liu et al. method [4] and the Zhang et al. method [5] were introduced as a comparison, and the stitching experiment was carried out. The test was carried out according to the accuracy and time. The results are as follows.

### 3.2. Analysis of Experimental Results

Table 1 shows the accuracy of three smooth stitching methods for the texture seams of remote sensing images.

**Table 1.** Accuracy of three smooth stitching methods for the texture seams of remote sensing images.

| Experiment Number | Accuracy of Stitching | | |
|---|---|---|---|
| | Liu et al. Method [4] | Zhang et al. Method [5] | Proposed Method |
| Experiment 1 | 75.60% | 86.40% | 98.50% |
| Experiment 2 | 65.10% | 75.80% | 90.60% |
| Experiment 3 | 98.60% | 94.80% | 98.10% |

From the results in Table 1, it can be seen that in the three experiments, the stitching accuracy by the Zhang et al. method [5] method was higher than the Liu et al. method [4], but lower than the proposed method. In order to ensure the accuracy of the experimental results, each group of remote sensing images was subjected to 10 experiments to obtain the average value of the smooth stitching accuracy. The results are shown in Figure 3.

It can be seen from the results in Figure 3 that with the increase in the number of remote sensing images, the average accuracy of the three methods was gradually decreasing, and the accuracy of the proposed method was significantly higher than that of the other two methods.

The running time of three smooth stitching methods are shown in Figure 4.

It can be seen from the results in Figure 4 that with the increase in the number of experiments, the running time of the Liu et al. method [4] increased significantly (almost 23–25%). When the number of experiments reached 100, the smooth stitching time of the texture seams of the remote sensing images was 27.8 and 29 s, respectively, and the

stitching time of the proposed method changed slowly. When the number of experiments reached 100, the smooth stitching time by the proposed method was 9 s, indicating that the proposed method could speed up the smooth stitching of the texture seams of remote sensing images, thereby improving the efficiency of smooth stitching of the texture seams of remote sensing images.

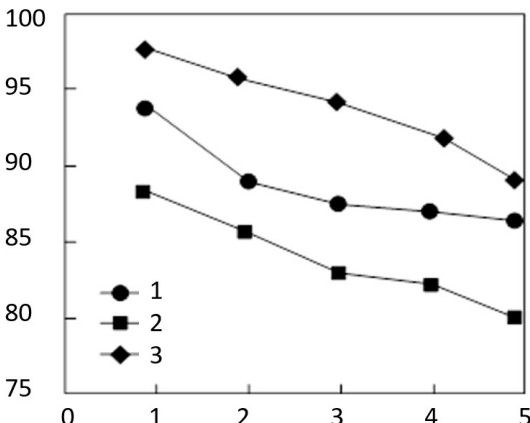

**Figure 3.** The average accuracy of the three smooth stitching methods.

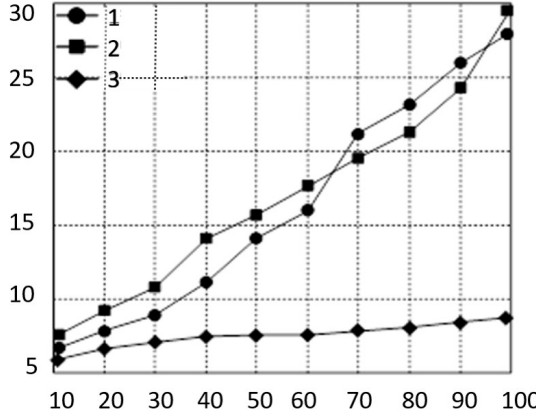

**Figure 4.** Running time of the three smooth stitching methods.

## 4. Conclusions

Taking into account the problems of the traditional smooth stitching methods for the texture seams of remote sensing images, the local texture overlapping area of remote sensing images was optimized and fused, and the optimal stitching line of the texture seams was extracted. By designing a smooth stitching method for the texture seams of remote sensing images, this research realized the smooth stitching of the texture seams of remote sensing images. The results show that the proposed method has great advantages in stitching effect and is suitable for widespread use. The test was carried out according to accuracy and time and the proposed method achieved better results by almost 25%.

On the other hand, the proposed image stitching method mainly solves the problem of improving registration accuracy by designing a good transformation function. For UAV remote sensing image mosaics, additional information is usually needed, such as camera calibration parameters, or position and rotation data obtained from the GPU/IMU, ground control points, or a reference map, to achieve accurate mosaic results. However, these additional parameters do not focus on the ghost phenomenon caused by moving objects, and the ghosts cannot be eliminated by a transform function. The smooth image stitching method can alleviate these problems to a certain extent. However, there are still limitations in dealing with ghosts due to the following reasons: first, when there is a moving object in the stitched image, the seam may cut or copy the object; secondly, the seam-driven method

selects one of the multiple seams as the final seam, and if there are fast-moving objects in the stitched images, the process of seam finding method becomes challenging.

In short, the proposed method cannot deal with all the challenges of eliminating ghosts, and ghost elimination is an urgent problem. In particular, when shooting images that are generally used for stitching, they may be at the same position or have a small rotation. However, when the UAV's bird's-eye view camera captures images, important factors are not only translation and rotation, but also the distance between the camera and the object, and the level of shaking, resulting in small objects in UAV remote sensing images and poor image quality, which in turn increases the difficulty of image stitching and the removal of ghosts.

In future work, an adaptive threshold adjustment method will be designed to select an appropriate threshold for each group of test images to improve the robustness of the method. Additionally, a method to stitch UAV remote sensing images while eliminating ghosts without a seam finding process will be investigated.

**Funding:** This research received no external funding.

**Institutional Review Board Statement:** Not applicable.

**Informed Consent Statement:** Not applicable.

**Conflicts of Interest:** The authors declare no conflict of interest.

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
