# Peer review of "Smooth Stitching Method for the Texture Seams of Remote Sensing Images Based on Gradient Structure Information"

_processes, doi:10.3390/pr9101689_

Round 1
Reviewer 1 Report
Dear Dr. Author.
Thank you for submitting your manuscript,
The authors prosed a smooth stitching method for the texture seams of remote sensing images based on gradient structure information.
Unfortunately, the paper is not well written and it takes some time to fully understand the methodology.
In general, this paper is not well written and the equations are not well written. For these reasons, I recommend to reconsider after major revision of the paper.
Author Response
I am thankful for the comprehensive review and the valuable suggestions to improve the quality of my work. The point-to-point response to the respected Reviewers is given as follows. The modifications in the manuscript according to the Reviewers comments are highlighted in red color. I deeply appreciate the time and effort you have spent in reviewing my manuscript.
Reviewer 1
The authors prosed a smooth stitching method for the texture seams of remote sensing images based on gradient structure information. Unfortunately, the paper is not well written and it takes some time to fully understand the methodology. In general, this paper is not well written and the equations are not well written. For these reasons, I recommend to reconsider after major revision of the paper.
I have revised (almost 30%) the manuscript taking into account all helpful comments exactly to improve the readability of the research paper. Also, I have re-arranged the entire paper and improved a lot on the background in order to make the paper self-consistent. I have explained thoroughly where the original contribution of the work is, the novelty of the contribution, and I have extended the experiments of the study’s arguments. Furthermore, I have given more details about the considered architecture and about the application of the proposed method to it. I think, with improvements that have been suggested by reviewers, the paper has improved significantly and makes an acceptable case for publication. I deeply appreciate the time and effort you have spent in reviewing my manuscript.
Reviewer 2 Report
The author needs to address the following issues: 1-In abstract, there are no numbers that shows the comparison of the approach against any other approaches in the LR. It is claimed that this is good?? How good is the approach???Line 295 and 296: “The method of literature [4] and the method of literature [5] are introduced as a comparison, and the stitching experiment is carried out.” Mention these in your abstract… 2-Table 1… change the name of literature 4 and 5 to specific names… 3- Conclusion doesn’t talk about any limitation of the work. In fact, there can be a lot, since it is just simulation of the experiment. 4- the references are very low in number. need to increase the LR.Author Response
I am thankful for the comprehensive review and the valuable suggestions to improve the quality of my work. The point-to-point response to the respected Reviewers is given as follows. The modifications in the manuscript according to the Reviewers comments are highlighted in red color. I deeply appreciate the time and effort you have spent in reviewing my manuscript.
Reviewer 2
The author needs to address the following issues:
1-In abstract, there are no numbers that shows the comparison of the approach against any other approaches in the LR. It is claimed that this is good?? How good is the approach??? Line 295 and 296: “The method of literature [4] and the method of literature [5] are introduced as a comparison, and the stitching experiment is carried out.” Mention these in your abstract…
The missing explanations were successfully added in the Abstract according to reviewer suggestions. Specifically, “Experimental results show that the design method has good performance in stitching accuracy and efficiency in the smooth stitching of the texture seams of remote sensing images. Specifically, the Liu et al. and the Zhang et al. methods that are the benchmark studies in the literature are introduced as a comparison, and the stitching experiment is carried out. The test is carried out according to accuracy and time and the proposed method achieves better almost 25%.”
2-Table 1… change the name of literature 4 and 5 to specific names…
The appropriate names - explanations were successfully added in Table 1 according to reviewer suggestions. Thank you for this constructive comment.
3- Conclusion doesn’t talk about any limitation of the work. In fact, there can be a lot, since it is just simulation of the experiment.
The appropriate missing explanations about limitation of the work were successfully added in Conclusion section. Specifically, “On the other hand, the proposed image stitching method mainly solves the problem of improving registration accuracy by designing a good transformation function. For UAV remote sensing image mosaics, additional information is usually needed, such as camera calibration parameters, or position and rotation data obtained from the GPU/IMU, ground control points, or a reference map, to achieve accurate mosaic results. However, these additional parameters do not focus on the ghost phenomenon caused by moving objects, and the ghosts cannot be eliminated by a transform function. The seam-driven image stitching method can alleviate these problems to a certain extent. However, there are still limitations in dealing with ghosts due to the following reasons: first, when there is a moving object in the stitched image, the seam may cut or copy the object; secondly, the seam-driven method selects one of the multiple seams as the final seam, and if there are fast-moving objects in the stitched images, the process of seam finding method becomes challenging. In short, the proposed method cannot deal with all the challenges of eliminating ghosts, and ghost elimination is an urgent problem. In particular, when shooting images that are generally used for stitching, they may be at the same position or have a small rotation. However, when the UAV’s bird’s-eye view camera captures images, important factors are not only translation and rotation, but also the distance between the camera and the object, and the level of shaking, resulting in small objects in UAV remote sensing images and poor image quality, which in turn increases the difficulty of image stitching and the removal of ghosts. In future work, an adaptive threshold adjustment method will be designed to select an appropriate threshold for each group of test images to improve the robustness of the method. Also, a method to stitch UAV remote sensing images while eliminating ghosts without a seam finding process will be investigated.” Thank you for the remarks.
4- the references are very low in number. need to increase the LR.
The LR significant improved as follow “Through various registration methods, the overlapping areas of two images can be well aligned, and the non-overlapping areas usually have serious distortions. Chang et al. [6] proposed (the SPHP) algorithm, which corrects the shape of the stitched image, and reduces the projection distortion. Lin et al. [7] proposed a homography linearization method, which is also a shape correction problem; the natural appearance of the stitching results is improved with this method by a level compared with SPHP. Chen et al. [8] proposed the global similarity prior (GSP) to obtain natural stitching results, which uses line alignment constraints to set constraint terms. Liao et al. [9] used two single-perspective warps to register images to produce natural stitching results. Li et al. [10] proposed a novel quasi-homography to solve the line blending problem between the homography transformation and the similarity transformation by linearly scaling the horizontal component of the homography, to create a more natural panorama. Also, UAV image mosaic usually needs additional information, such as camera calibration parameters, and position and rotation data from the GPS/IMU, ground control points or a reference map, to achieve accurate mosaic results. Xue et al. [11] proposed a novel image mosaic method without the requirements of camera calibration parameters, camera poses, or any 3D reconstruction procedure. This method can obtain visually satisfactory mosaic results automatically using only the raw 2D imagery captured from a UAV. Liu et al. [12] considered mosaic for low overlap UAV aerial images and found that the commonly used methods and software were not suitable for this type of data. Then, a novel adjustment model for mosaic low-overlap sweeping images was proposed. Cai et al. [13] proposed a robust transform estimator based on residual analysis and its application to UAV aerial images. Estimating the transformation between two images from the same scene is a fundamental step for image registration, image stitching and 3D reconstruction”. Thank you for your careful reading.